# When to use commuting zones? An empirical description of spatial autocorrelation in U.S. counties versus commuting zones

Craig Wesley Carpenter[1,2]*, Michael C. Lotspeich-Yadao[3], Charles M. Tolbert[4]

1 Department of Agricultural, Food, and Resource Economics, Michigan State University, East Lansing, Michigan, United States of America, 2 Department of Agricultural Economics, Texas A&M University, College Station, Texas, United States of America, 3 College of Applied Health Sciences, University of Illinois at Urbana-Champaign, Urbana, Illinois, United States of America, 4 Department of Sociology, Baylor University, Waco, Texas, United States of America

* cwcarp@msu.edu

**Data Availability Statement:** All data were collected at the county level in 2000, which is the last year that the Decennial Census included a long-form questionnaire that provided journey-to-

## Abstract

U.S. Commuting Zones (CZs) are an aggregation of county-level data that researchers commonly use to create less arbitrary spatial entities and to reduce spatial autocorrelation. However, by further aggregating data, researchers lose point data and the associated detail. Thus, the choice between using counties or CZs often remains subjective with insufficient empirical evidence guiding researchers in the choice. This article categorizes regional data as entrepreneurial, economic, social, demographic, or industrial and tests for the existence of local spatial autocorrelation in county and CZ data. We find CZs often reduce—but do not eliminate and can even increase—spatial autocorrelation for variables across categories. We then test the potential for regional variation in spatial autocorrelation with a series of maps and find variation based on the variable of interest. We conclude that the use of CZs does not eliminate the need to test for spatial autocorrection, but CZs may be useful for reducing spatial autocorrelation in many cases.

## Introduction

Commuting Zones (CZs) are an aggregation of county-level data that researchers commonly used to create less arbitrary spatial entities and to reduce spatial autocorrelation. Researchers use commuting zones extensively in high-impact articles and research projects across disciplines [1–5]. However, by further aggregating data, researchers lose observations and the associated detail. Thus, without empirical evidence for guidance, researchers often subjectively choose between counties or CZs.

To shed further light on this issue, we use the Moran's I and Geary's c statistics to test for overall spatial dependence in regional industrial, economic, social, and demographic indicators. At this level, we find CZs reduce—but do not eliminate—spatial autoregression for most

work information. This data is all publicly available online (https://data.census.gov/cedsci/). Commuting Zone crosswalks are also publicly available online (https://www.ers.usda.gov/data-products/commuting-zones-and-labor-market-areas/).

**Funding:** This project was supported by the Agricultural and Food Research Initiative Competitive Program of the USDA National Institute of Food and Agriculture (NIFA, https://nifa.usda.gov/), award numbers 2017-67023-26242 (CC) and 2018-68006-34968 (CC, ML, CT). NIFA funded the time and data collection for the coauthors.

**Competing interests:** The authors have declared that no competing interests exist.

variables. We then test for spatial heterogeneity with the Local Moran's I tool [6–8] and find expected variation based on the variable of interest.

We proceed with background on the development of commuting zones and their use relative to counties. Then, we provide an overview of the spatial econometric methods to be used, categorize a selection of ecological measures as industrial, economic, social, or demographic data, and test for the extent of spatial autocorrelation, comparing results between counties and CZs. We conclude with a simple empirical guide to help researchers justify their choice between counties and CZs when spatial autocorrelation is a potential concern.

## Commuting zones background

There has long been an interest in local labor markets [9–11]. The local labor market is viewed as a set of relationships between employers and workers. These relationships exist in a space bounded by places of work and residence. Before Commuting Zones (CZs), some researchers [12–14] relied exclusively on metropolitan area definitions such as Standard Metropolitan Statistical Areas (SMSAs) and Metropolitan Statistical Areas (MSAs). Studies employing these measures of labor market areas excluded nonmetropolitan places by definition. [15, 16] developed an inclusive county group scheme known as the Bureau of Economic Analysis (BEA) areas. Based on central place theory, commuting nodes are identified and surrounding counties are assigned to nodes based on commuting patterns. Because BEA areas and SMSAs tends to have an urban center, these schemas were particularly troublesome for measuring of rural labor markets.

Of course, one could simply use counties themselves. However, drawbacks in the use of counties as labor market areas is that the geography is administrative and determined independently by state officials (so criteria vary by state) and counties do not cross state lines. Individuals may cross county and state lines often as part of their regular commute. All these drawbacks could distort measures of labor market areas. We note the geography literature related to the Modifiable Areal Unit Problem (MAUP), noting that summarized data can be distorted based on the spatial units used, overlaps with our discussion herein. See, e.g., [17] and [18] for related discussions.

Thus, [19] developed Commuting Zones (CZs). Their approach addressed the aforementioned drawbacks of alternatives by including all U.S. counties (and county equivalents), used uniform criteria for designating labor market areas, and did not require each area to have an urban center. Specifically, they measured county-to-county flows of commuters with a hierarchical cluster algorithm to group counties with strong commuting ties. In their analysis of 1990 data, 741 CZs were delineated for all U.S. counties and county equivalents. Mirroring their original notation, first, proportional flow measures were defined as

$$P_{ij} = P_{ji} = \frac{f_{ij} + f_{ji}}{\min\left(rlf_i, \ rlf_j\right)}$$

where $f_{ij}$ is the number of commuters from county $i$ to county $j$, $f_{ji}$ is the number of commuters from county $j$ to county $i$, $rlf_i$ is the resident labor force in county $i$, $rlf_j$ is the resident labor force in county $j$. The main diagonal of the flow matrices was set to zero ($P_{ij} = P_{ji} = 0$ if $i = j$). (See [20, 21] for discussions of the tradeoffs in this approach.) So, a symmetric matrix of $P_{ij}$ would be characterized as a similarity matrix with greater commuting relationships

implying higher values of $P_{ij}$. But rather than similarity measures, the clustering algorithm required a matrix of distance coefficients (i.e., a dissimilarity matrix), so [19] expressed the proportional flow measures as distance measures:

$$D_{ij} = D_{ji} = 1 - P_{ij}$$

[19] then used hierarchical cluster analysis to develop the commuting zones. They found that clusters forming at or above the 0.98 level are considered sufficiently distant from one another to warrant separation. This paper uses CZs defined by the same method with Census 2000 journey-to-work data from the long-form questionnaire [22].

## Modifiable areal unit problem

The Modifiable Areal Unit Problem (MAUP) is a challenge in spatial regression modeling; a) results can vary based on the aggregation of individual points into larger units [scale], and b) results can vary by how these larger units are defined, leading to ecological fallacies [aggregation] [23, 24]. As a result, the MAUP is a foreboding concern facing spatial economists as an early decision in geographic areal units can alter the outcome of the multivariate analysis [25]. It has been well-established in the literature that analytical results are sensitive to MAUP [26]. [27] for instance demonstrate the sensitivity of the U.S. CZ definitions to two features of the CZ methodology: the count of clusters and uncertainty in the input data. Both important concerns relate to CZ delineations more generally, while this article focuses on the specific CZ delineations commonly used by researchers. Uncertainly of the input data related to the margins of error implicit in the journey-to-work data used by researchers to calculate commuting flows in the U.S. This latter concern is more acute for CZ delineations developed from years after 2000 in which the journey-to-work data relied on the (smaller sample) American Community Survey. As we discuss below in the data section, this article uses CZs developed from the 2000 long-form U.S. Census, which had a larger sample (and smaller margins of error). Nonetheless, [27] points are particularly important for researchers applying [19] methodology to develop new labor market areas outside the U.S.

One potential way to address the MAUP, according to Openshaw [23], is to identify and utilize a problem-specific approach to select an optimal system of zoning. By activating prior knowledge in the operationalization process, researchers are less likely to overlook an ecological fallacy of aggregation. Past efforts have been made to educate researchers in child welfare [28, 29], public health [30], and political science [31]. We believe the next step is to offer an exploratory narrative to help researchers purposefully select between U.S. counties and commuting zones for social science research.

## Methods

Naturally, many researchers argue that using CZs eliminates the need to include commuting in the analysis, since the boundaries of these areas are drawn to minimize commuting flows. But CZs provide only an imperfect measure of local labor markets, with heterogeneous—but substantial—commuting across CZ boundaries, emphasizing the potential need for the inclusion of the share of residents who work locally (or who commute) to be included in reduced-form regressions to help to control for this heterogeneity in commuting patterns across CZ boundaries [32]. ([32] find that for the median CZ, 7 percent of its residents work outside the

CZ and 7 percent of its workers live outside the CZ; they further show that at the ninety-fifth percentile, these figures rise to 22 percent and 15 percent, respectively.) Two tests can be used to test for the strength and significance of spatial dependence: the Moran's I statistic (global), and Geary's c statistic (global).

The formula of Moran's I (global) is given by

$$I = \frac{N}{\sum_{i=1}^{n}\sum_{j=1}^{n} w_{ij}} \frac{\sum_{i=1}^{n}\sum_{j=1}^{n} w_{ij}(x_i - \bar{x})\left(x_j - \bar{x}\right)}{\sum_{i=1}^{n}(x_i - \bar{x})^2}$$

where $(x_i - \bar{x})$ is the deviation of a variable for county or CZ $i \neq j$ from its mean and $w_{ij}$ is the $ij$th element of the spatial weight matrix $W$ [33, 34]. The Global Moran's I lies within the range $[-1, 1]$ when the $W$ is row-standardized and a positive or negative value of Moran's I indicates positive or negative spatial autocorrelation.

A Moran scatter plot illustrates a spatial autocorrelation in standardized (z-score) variables and the standardized spatial lags of the variables to help determine is the results are driven by outliers [6]. Moran scatter plots for all variables are available in the Supporting Information (S1–S4 Figs).

Another test of global autocorrelation is the Geary's c (global) statistic, which is more sensitive to local autocorrelation than the Moran's I (global) [35]. The formula for Geary's c (global) is

$$c = \left(\frac{n-1}{2\sum_{i=1}^{n}\sum_{j=1}^{n} w_{ij}}\right) \frac{\sum_{i=1}^{n}\sum_{j=1}^{n} w_{ij}\left(y_i - y_j\right)^2}{\sum_{i=1}^{n}(y_i - \bar{y})^2}$$

where $(y_i - \bar{y})$ is the deviation of a variable for county or CZ $i \neq j$ from its mean and $w_{ij}$ is the $ij$th element of the spatial weight matrix $W$ [36, 37]. The Geary's c statistic is always positive and ranges from 0 to 2; when positively correlated, the statistic is less than 1 [8]. When negatively correlated, c will be greater than 1 [38].

## Regional variation in spatial autocorrelation

While the Global Moran's I and Geary's c statistics return measures of overall spatial dependence, we expect spatial variation in spatial autocorrelation and thus use the Local Moran's I tests for regional structures of spatial autocorrelation. The formula for Moran's I (local) is

$$I_i = (y_i - \bar{y}) \sum_{j=1, j\neq i}^{n} w_{ij}\left(y_j - \bar{y}\right)$$

which is a local version of the Moran's $I$ statistic defined previously but takes into consideration the relationship between $y_i$ and neighboring observations based on the weight matrix [6, 36]. We use the contiguity-based spatial weighting matrix; this is because there is a great variation in the area of counties and CZs across the United States that would hinder the use of weights based on Euclidian Distance.

Regional data, such as the concentration of industry or racial composition, may span to create spatial clusters of high values which contribute proportionally more to the global indicator [6]. Past efforts to identify these spatial clusters have included point pattern analysis logic of the Local G Statistic [39, 40] and [6, 33] Moran Scatterplot and Local Indicators of Spatial

Association (LISA). Tests for spatial heterogeneity in the unit selection process have been documented by [41] and [42].

The local cluster map, included for all variables in the Supporting Information S5–S9 Figs, shows how spatial clusters and outliers arise for both counties and CZs based on the concentration of these high/low values. A cluster of counties or CZs with high values are considered a 'High-High cluster', whereas a singular county or CZ with a statistically significant high value compared to neighbors is considered a 'High-Low outlier.' Similarly, a cluster of low values is a 'Low-Low cluster' and a low outlier is a 'Low-High outlier.'

## Data

All data were collected at the county level in 2000, which is the last year that the Decennial Census included a long-form questionnaire that provided journey-to-work information. The Census 2000 shapefile was obtained from the U.S. Census Bureau to represent county-level geography [43] and the Commuting Zones shapefile from [44]. The commuting data were generally collected from one out of every six U.S. housing units. The sampling factor for rural housing units was one of two, yielding a robust sample from which the county-to-county commuting flows could be calculated. Although more recent delineations exist, these rely on smaller samples based on the American Community Survey. We have reservations about the robustness of these newer delineations, especially for rural areas. Regardless, year-to-year

**Table 1. Data summary and literature.**

| Ecological domain | Measure/variable | Source | Literature |
|---|---|---|---|
| Industrial | LQs for each 2-digit NAICS | CBP | [45] |
| Entrepreneurial | % workforce self-employed | DC | [46] |
| | Businesses with 1–4 employees | CBP | [47] |
| | % creative class | DC, SF 4 | [48] |
| Economic | Total bank deposits | FDIC | [49] |
| | % population below poverty | SAIPE | [50] |
| | Unemployment rate | CPS, BLS | [51] |
| | Per capita income | DC | [52] |
| Social | Small manufacturing per 10,000 | CBP | [53] |
| | Associations per 10,000 | CBP | [54] |
| | Third places per 10,000 | CBP | [54] |
| | Voters in 2000 | CQ Press | [2] |
| | Adherents to civic denominations | RCMS | [53] |
| Demographic | % population identify as Black | DC | [55] |
| | % population identify as Hispanic | DC | [56, 57] |
| | % adult population with ≥ bachelor's | DC | [58] |
| | % population age 25 and younger | PEP | [59] |
| | % population age 65 and older | DC | [59] |

*Notes*: Table summarizes the example variables included in this article by ecological domain, its source, and related literature. We impute suppressed values of the CBP using [60], which [61] show reduces measurement error over common alternatives. All data is from the year 2000. Literature column gives example literature indicating the relevance of each variable.

*Abbreviations*: RCMS, Religious Congregations and Membership Study, 2000; BLS, Bureau of Labor Statistics; CBP, County Business Patterns; CPS, Current Population Survey; CQ Press, Voting and Election Collection; DC, Decennial Census; FDIC, Federal Deposit Insurance Corporation; LQs, location quotient; NAICS, North American Industrial Classification System; PEP, U.S. Census Bureau's Population Estimates Program; SAIPE, Small Area Income and Poverty Estimates; SF, Summary File.

changes in the CZ delineations remain small. We argue that the external validity of the results in this article likely holds for other CZ years/delineations [27].

We test data across five ecological domains: industrial, entrepreneurial, economic, social, and demographic. Table 1 summarizes each variable, source, justification, and associated literature.

## Results

Tables 2 and 3 show the significance of the Moran's I for variables when using counties compared to when using CZs. In general, the Moran's I significance indicates that, while CZs reduce the amount of spatial autocorrelation in many of the variables across the ecological domains under consideration, they do not eliminate it. Additionally, in some cases spatial autocorrelation even slightly increases.

In general, the Moran scatterplots, which are available in the Supporting Information, indicate, first, that the results are not driven by outliers. Second, consistent with the findings from Moran's I significance tests, although CZs weaken spatial autocorrelation in many cases, they do not eliminate it in almost all cases and even increase spatial autocorrelation in some cases.

**Table 2. Global Moran's I for entrepreneurial, economic, social, and demographic domains (Counties vs. CZs).**

| Ecological domain | Measure/variable | Counties | | | CZs | | |
|---|---|---|---|---|---|---|---|
| | | Moran's I | std. err. | z-score | Moran's I | std. err. | z-score |
| Entrepreneurial | % workforce self-employed | 0.603*** | .011 | 56.063 | 0.476*** | .023 | 20.636 |
| | Businesses with 1–4 employees | 0.267*** | .011 | 24.847 | 0.398*** | .023 | 17.226 |
| | % creative class | 0.472*** | .011 | 43.809 | 0.258*** | .023 | 11.188 |
| Economic | Total bank deposits | 0.166*** | .010 | 15.887 | 0.348*** | .023 | 15.517 |
| | % population below poverty | 0.612*** | .011 | 56.831 | 0.535*** | .023 | 23.197 |
| | Unemployment rate | 0.444*** | .011 | 41.285 | 0.392*** | .023 | 17.064 |
| | Per capita income | 0.564*** | .011 | 52.429 | 0.412*** | .023 | 17.835 |
| Social | Associations per 10,000 | 0.459*** | .011 | 42.622 | 0.513*** | .023 | 22.262 |
| | Third places per 10,000 | 0.368*** | .011 | 34.376 | 0.368*** | .023 | 16.090 |
| | Voters in 2000 | 0.425*** | .011 | 40.083 | 0.602*** | .023 | 26.012 |
| | Adherents to civic denominations | 0.744*** | .011 | 69.051 | 0.747*** | .023 | 32.267 |
| Demographic | % population identify as Black | 0.790*** | .011 | 73.426 | 0.837*** | .023 | 36.250 |
| | % population identify as Hispanic | 0.825*** | .011 | 76.820 | 0.807*** | .023 | 35.182 |
| | % adult population with $\geq$ bachelor's | 0.411*** | .011 | 38.207 | 0.225*** | .023 | 9.751 |
| | % population age 25 and younger | 0.333*** | .011 | 30.953 | 0.274*** | .023 | 11.921 |
| | % population age 65 and older | 0.502*** | .011 | 46.616 | 0.338*** | .023 | 14.637 |

Significance levels:

***<1%,

**<5%,

*<10%

*Notes*: Table summarizes the Global Moran's I Test for Spatial Dependence. This table uses a contiguity-based spatial weighting matrix to produce the Moran's I Index (displayed), Expected Index, Variance (displayed), z-score (displayed), and p-value (displayed). The weight was constructed in GeoDa 1.16, and test performed with the *SPATGSA* [62] command in Stata 17/SE. Significance of the test when using an inverse-distance based spatial matrix and a joint test of whether either of the contiguity or inverse distance matrices specify a spatial dependence are provided in the (S2–S5 Tables).

*Abbreviations* Voters in 2000, percent of eligible population that vote in presidential election (2000) per 1000 population; Total bank deposits, commercial banks and savings institutions—total deposits; std err, standard error.

Table 3. Global Moran's I for industrial LQs (Counties versus CZs).

| Ecological domain | Measure/variable | Counties | | | CZs | | |
|---|---|---|---|---|---|---|---|
| | | Moran's I | std. err. | z-score | Moran's I | std. err. | z-score |
| Industrial LQs | 11: Ag, Forestry, etc. | 0.274*** | .011 | 25.738 | 0.305*** | .023 | 13.571 |
| | 21: Mining, Quarrying, Oil, etc. | 0.319*** | .011 | 29.952 | 0.268*** | .023 | 11.858 |
| | 22: Utilities | 0.048*** | .011 | 4.536 | 0.093*** | .023 | 4.195 |
| | 23: Construction | 0.146*** | .011 | 13.614 | 0.199*** | .023 | 8.750 |
| | 31–33: Manufacturing | 0.429*** | .011 | 39.800 | 0.467*** | .023 | 20.194 |
| | 42: Wholesale Trade | 0.220*** | .011 | 20.510 | 0.259*** | .022 | 11.962 |
| | 44–45: Retail Trade | 0.150*** | .011 | 13.963 | 0.272*** | .023 | 11.867 |
| | 48–49: Transportation and Warehousing | 0.049*** | .010 | 4.664 | 0.065*** | .023 | 2.900 |
| | 51: Information | 0.042*** | .011 | 3.948 | 0.078** | .023 | 3.470 |
| | 52: Finance and Insurance | 0.143*** | .011 | 13.453 | 0.135*** | .023 | 5.959 |
| | 53: Real Estate and Rental and Leasing | 0.110*** | .010 | 10.888 | 0.212*** | .023 | 9.335 |
| | 54: Prof., Scientific, and Tech. Services | 0.252*** | .011 | 23.684 | 0.162*** | .023 | 7.090 |
| | 55: Management of Enterprises | 0.102*** | .011 | 9.563 | 0.106*** | .023 | 4.696 |
| | 56: Admin., Support, Waste Management, etc. | 0.140*** | .011 | 13.049 | 0.219*** | .023 | 9.670 |
| | 61: Educational Services | 0.048*** | .011 | 4.494 | 0.121*** | .022 | 5.641 |
| | 62: Health Care and Social Assistance | 0.175*** | .011 | 16.306 | 0.338*** | .023 | 14.661 |
| | 71: Arts, Entertainment, and Recreation | 0.113*** | .010 | 10.916 | 0.085*** | .022 | 3.916 |
| | 72: Accommodation and Food Services | 0.220*** | .011 | 20.557 | 0.307*** | .023 | 13.395 |
| | 81: Other Services | 0.167*** | .011 | 15.576 | 0.128*** | .023 | 5.666 |
| Observations | | 3,109 | | | 691 | | |

Significance levels:

***<1%,

**<5%,

*<10%

*Notes*: Table summarizes the Global Moran's I Test for Spatial Dependence. This table uses a contiguity-based spatial weighting matrix to produce the Moran's I Index (displayed), Expected Index, Variance (displayed), z-score (displayed), and p-value (displayed). The weight was constructed in GeoDa 1.16, and test performed with the *SPATGSA* [62] command in Stata 17/SE. Significance of the test when using an inverse-distance based spatial matrix and a joint test of whether either of the contiguity or inverse distance matrices specify a spatial dependence are provided in the (S2–S5 Tables).

*Abbreviations*: LQs, Location Quotients; NAICS, North American Industrial Classification System; std err, standard error.

Broad cultural, historical, and natural structures, such as that of racial stratification or extractive industry, will continue to drive the global indicator.

To demonstrate, we use the Moran's *I* (local) statistic and LISA to visualize the potential for regional variation in spatial autocorrelation in the Supporting Information. Example results, presented in Fig 1, indicate that results are not generally driven by regional outliers and rather, the spatial unit selected can alter the statistical significance of the clusters. When taking into consideration manufacturing, for instance, low concentration clusters appear when using CZs but disappear from the west coast when using counties as the spatial unit. For the percent of residents identifying as Black, Appalachia as a whole is a Low-Low cluster at the county-level but a Low-High outlier to neighboring units when CZs are used. This is a clear example of the MAUP, as the statistical significance of the Moran's *I* (local) statistic can change based on the spatial units used. While CZs may change local regional concentration of spatial autocorrelation, larger regional variation found with counties is not typically eliminated. Full results are available in the Supporting Information.

(A)

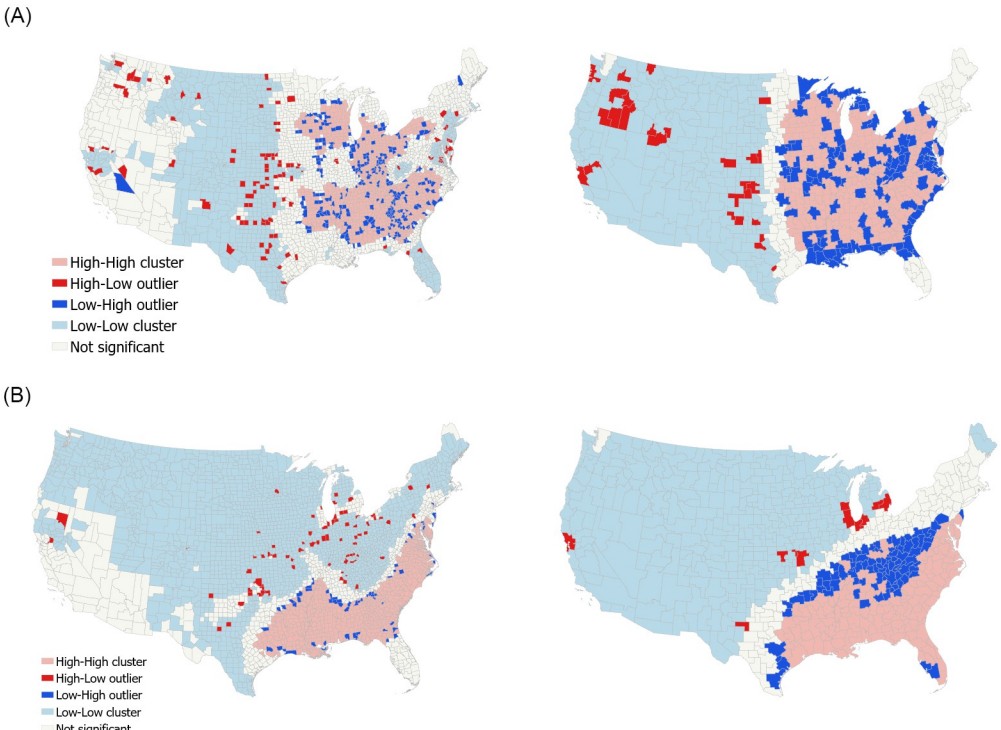

(B)

**Fig 1.** a. LISA Cluster Map for Spatial Autocorrelation (counties left, CZs right); NAICS 31–33 (Manufacturing). b. LISA Cluster Map for Spatial Autocorrelation (counties left, CZs right); Percent identifying as Black *Note*: Shapefiles republished from [44] under a CC BY license, with permission from Christopher S. Fowler, original copyright 2020.

Finally, local spatial autocorrelation may also be a concern and Geary's c, while still a global measure is more sensitive to local autocorrelation than the Global Moran's I [35], so we provide the Geary's c statistics in Tables 4 and 5. In general, we find results consistent with Moran's I, indicating positive spatial autocorrelation across ecological domains.

## Summary and conclusions

There are additional considerations for researchers beyond the potential for spatial autocorrelation. First, researchers are often interested in the effects of policy interventions at spatial scales smaller than CZs. Second, researchers should consider the time of their analysis relative the potential for differential commuting patterns; commuting patterns changed substantially 1960–2020 [32]. And third, it is important to note that, depending on the geography of interest, commuting between CZs can still exist in substantial quantities. Of course, CZs combine the more-significant counties, but substantial commuting can still occur between CZs. Notably, in [19] note in their original work, emphasized that CZs "were principally intended to be statistical units for analysis of nonmetropolitan labor market performance and employment problems."

There is also valuable research that applies the [19] methodology to different countries and datasets or for characterizing labor markets outside the United States [63–66]. In addition to concerns related to cluster count subjectivity and input data margins of error [27],

**Table 4. Global Geary's c for entrepreneurial, economic, social, and demographic domains (Counties vs. CZs).**

| Ecological domain | Measure/variable | Counties | | | CZs | | |
|---|---|---|---|---|---|---|---|
| | | Geary's c | std. err. | z-score | Geary's c | std. err. | z-score |
| Entrepreneurial | % workforce self-employed | 0.390*** | .013 | -47.435 | 0.481*** | .028 | -18.731 |
| | Businesses with 1–4 employees | 0.719*** | .012 | -22.609 | 0.561*** | .026 | -17.184 |
| | % creative class | 0.526*** | .012 | -38.599 | 0.768*** | .025 | -9.230 |
| Economic | Total bank deposits | 0.789*** | .037 | -5.725 | 0.721*** | .045 | -6.265 |
| | % population below poverty | 0.382*** | .012 | -50.739 | 0.457*** | .028 | -19.501 |
| | Unemployment rate | 0.537*** | .013 | -34.844 | 0.582*** | .029 | -14.287 |
| | Per capita income | 0.431*** | .013 | -44.075 | 0.576*** | .026 | -16.145 |
| Social | Associations per 10,000 | 0.511*** | .012 | -39.378 | 0.444*** | .028 | -19.748 |
| | Third places per 10,000 | 0.592*** | .018 | -22.406 | 0.535*** | .033 | -13.927 |
| | Voters in 2000 | 0.546*** | .027 | -16.851 | 0.366*** | .025 | -25.746 |
| | Adherents to civic denominations | 0.252*** | .011 | -65.023 | 0.237*** | .025 | -30.322 |
| Demographic | % population identify as Black | 0.204*** | .013 | -62.830 | 0.175*** | .027 | -30.099 |
| | % population identify as Hispanic | 0.171*** | .016 | -53.037 | 0.187*** | .033 | -24.955 |
| | % adult population with ≥ bachelor's | 0.585*** | .013 | -33.182 | 0.767*** | .026 | -9.016 |
| | % population age 25 and younger | 0.637*** | .012 | -29.096 | 0.678*** | .027 | -11.829 |
| | % population age 65 and older | 0.490*** | .012 | -43.771 | 0.648*** | .025 | -13.946 |
| Observations | | 3,109 | | | 691 | | |

Significance levels:

***<1%,

**<5%,

*<10%

*Notes*: Table summarizes the Global Geary's c Test for Spatial Autocorrelation. This table uses a contiguity-based spatial weighting matrix to produce Geary's contiguity ratio [0,2] (displayed), Variance (displayed), z-score (displayed), and p-value (displayed). The weight was constructed in GeoDa 1.16, and test performed with the *SPATGSA* [62] command in Stata 17/SE. Significance of the test when using an inverse-distance based spatial matrix and a joint test of whether either of the contiguity or inverse distance matrices specify a spatial dependence are provided in the (S2 and S3 Tables).

*Abbreviations*: Voters in 2000, percent of eligible population that vote in presidential election (2000) per 1000 population; Total bank deposits, commercial banks and savings institutions—total deposits; std. err., standard error.

many of the assessed variables of interest may differently cluster in a different country context. For example, commuting patterns themselves can differ vastly among high-income countries with different levels of investment in public transportation. Future research may include applying similar analyses as those herein to other countries (and their labor market areas).

In sum, researchers cannot dismiss the potential for spatial autocorrelation when using CZs. CZs do indeed reduce the severity of spatial autocorrelation in many cases, but researchers cannot use CZs as a route to ignore spatial autocorrelation completely because in some cases CZs may even increase spatial autocorrelation. Thus, if spatial unit aggregation is not fundamentally limiting, using CZs as spatial units (rather than counties) may be preferable for research in which spatial autocorrelation is a concern, but researchers should still test for spatial autocorrelation with both CZs and counties for the variables involved in their specific research question. The finding that CZs may be preferable is particularly salient when researchers examine topics related to (or potentially influenced by) local transportation.

Table 5. Global Geary's c for industrial LQs (Counties versus CZs).

| Ecological domain | Measure/variable | Counties | | | CZs | | |
|---|---|---|---|---|---|---|---|
| | | Geary's c | std. err. | z-score | Geary's c | std. err. | z-score |
| Industrial LQs | 11: Ag, Forestry, etc. | 0.693*** | .024 | -12.749 | 0.596*** | .043 | -9.392 |
| | 21: Mining, Quarrying, Oil, etc. | 0.633*** | .024 | -15.165 | 0.744*** | .040 | -6.433 |
| | 22: Utilities | 0.889*** | .032 | -3.516 | 0.855*** | .043 | -3.410 |
| | 23: Construction | 0.824*** | .017 | -10.240 | 0.689*** | .035 | -8.859 |
| | 31–33: Manufacturing | 0.556*** | .011 | -38.996 | 0.502*** | .024 | -20.358 |
| | 42: Wholesale Trade | 0.758*** | .016 | -15.482 | 0.689*** | .059 | -5.284 |
| | 44–45: Retail Trade | 0.834*** | .013 | -13.224 | 0.703*** | .031 | -9.701 |
| | 48–49: Transportation and Warehousing | 0.954* | .036 | -1.304 | 0.899*** | .039 | -2.564 |
| | 51: Information | 0.949** | .023 | -2.257 | 0.900*** | .034 | -2.968 |
| | 52: Finance and Insurance | 0.866*** | .024 | -5.514 | 0.892*** | .036 | -3.024 |
| | 53: Real Estate and Rental and Leasing | 0.844*** | .049 | -3.202 | 0.717*** | .036 | -7.791 |
| | 54: Prof., Scientific, and Tech. Services | 0.742*** | .025 | -10.482 | 0.881*** | .030 | -3.988 |
| | 55: Management of Enterprises | 0.907*** | .023 | -4.047 | 0.935** | .035 | -1.866 |
| | 56: Admin., Support, Waste Management, etc. | 0.843*** | .018 | -8.476 | 0.743*** | .037 | -6.890 |
| | 61: Educational Services | 0.953** | .023 | -2.033 | 0.824*** | .060 | -2.929 |
| | 62: Health Care and Social Assistance | 0.809*** | .013 | -14.858 | 0.632*** | .027 | -13.834 |
| | 71: Arts, Entertainment, and Recreation | 0.831*** | .040 | -4.246 | 0.866*** | .052 | -2.582 |
| | 72: Accommodation and Food Services | 0.771*** | .017 | -13.311 | 0.699*** | .032 | -9.466 |
| | 81: Other Services | 0.815*** | .015 | -12.146 | 0.864*** | .037 | -3.712 |
| Observations | | 3,109 | | | 691 | | |

Significance levels:

***<1%,

**<5%,

*<10%

*Notes*: Table summarizes the Global Geary's c Test for Spatial Autocorrelation. This table uses a contiguity-based spatial weighting matrix to produce Geary's contiguity ratio [0,2] (displayed), Variance (displayed), z-score (displayed), and p-value (displayed). The weight was constructed in GeoDa 1.16, and test performed with the *SPATGSA* [62] command in Stata 17/SE. Significance of the test when using an inverse-distance based spatial matrix and a joint test of whether either of the contiguity or inverse distance matrices specify a spatial dependence are provided in the Supporting Information (S2–S5 Tables).

*Abbreviations*: LQs, Location Quotients; NAICS, North American Industrial Classification System; std. err., standard error.

## Supporting information

**S1 Table. Additional descriptive statistics in Counties and CZs.** Significance levels: ***<1%, **<5%, *<10% *Notes*: Table provides the descriptive statistics for both counties and the CZs values. The numbers by the industries represent their two-digit NAICS code classification. *Abbreviations*: LQs, Location Quotients; NAICS, North American Industrial Classification System; Voters in 2000, percent of eligible population that vote in presidential election (2000) per 1000 population; Total bank deposits, commercial banks and savings institutions—total deposits (thousands); std. dev., standard deviation.
(DOCX)

**S2 Table. Global Moran's I for entrepreneurial, economic, social, and demographic domains (Counties versus CZs), inverse-distance spatial matrix.** Significance levels: ***<1%, **<5%, *<10% *Notes*: Table summarizes the Global Moran's I Test for Spatial Dependence. This table uses an inverse-distance based spatial weighting matrix to produce the Moran's I Index (displayed), Expected Index, Variance (displayed), z-score (displayed), and p-

value (displayed). The weight was constructed in GeoDa 1.16, and test performed with the SPATGSA [62] command in Stata 17/SE. *Abbreviations*: Voters in 2000, percent of eligible population that vote in presidential election (2000) per 1000 population; Total bank deposits, commercial banks and savings institutions—total deposits; std. err., standard error. (DOCX)

**S3 Table. Global Moran's I for industrial LQs (Counties versus CZs), inverse-distance spatial matrix.** Significance levels: ***<1%, **<5%, *<10% *Notes*: Table summarizes the Global Moran's I Test for Spatial Dependence. This table uses an inverse-distance based spatial weighting matrix to produce the Moran's I Index (displayed), Expected Index, Variance (displayed), z-score (displayed), and p-value (displayed). The weight was constructed in GeoDa 1.16, and test performed with the SPATGSA [62] command in Stata 17/SE. *Abbreviations*: LQs, Location Quotients; NAICS, North American Industrial Classification System; std. err., standard error. (DOCX)

**S4 Table. Global Geary's c for entrepreneurial, economic, social, and demographic domains (Counties versus CZs), inverse-distance spatial matrix.** Significance levels: ***<1%, **<5%, *<10% *Notes*: Table summarizes the Global Geary's c Test for Spatial Auto-correlation. This table uses an inverse-distance based spatial weighting matrix to produce Geary's contiguity ratio [0,2] (displayed), Variance (displayed), z-score (displayed), and p-value (displayed). The weight was constructed in GeoDa 1.16, and test performed with the SPATGSA [62] command in Stata 17/SE. *Abbreviations*: Voters in 2000, percent of eligible population that vote in presidential election (2000) per 1000 population; Total bank deposits, commercial banks and savings institutions—total deposits; std. err., standard error. (DOCX)

**S5 Table. Global Geary's c for industrial LQs (Counties versus CZs), inverse-distance spatial matrix.** Significance levels: ***<1%, **<5%, *<10% *Notes*: Table summarizes the Global Geary's c Test for Spatial Autocorrelation. This table uses an inverse-distance based spatial weighting matrix to produce Geary's contiguity ratio [0,2] (displayed), Variance (displayed), z-score (displayed), and p-value (displayed). The weight was constructed in GeoDa 1.16, and test performed with the SPATGSA [62] command in Stata 17/SE. *Abbreviations*: LQs, Location Quotients; NAICS, North American Industrial Classification System; std. err., standard error. (DOCX)

**S1 Fig. Moran scatter plot for entrepreneurship variables (counties left, CZs right).** (PDF)

**S2 Fig. Moran scatter plot for economic and social variables (counties left, CZs right).** (PDF)

**S3 Fig. Moran scatter plot for social variables (counties left, CZs right).** (PDF)

**S4 Fig. Moran scatter plot for demographic variables (counties left, CZs right).** (PDF)

**S5 Fig. LISA cluster map for industrial LQs (counties left, CZs right).** (PDF)

**S6 Fig. LISA cluster map for entrepreneurship variables (counties left, CZs right).** (PDF)

**S7 Fig. LISA cluster map for economic variables (counties left, CZs right).**
(PDF)

**S8 Fig. LISA cluster map for social variables (counties left, CZs right).**
(PDF)

**S9 Fig. LISA cluster map for demographic variables (counties left, CZs right).**
(PDF)

## Author Contributions

**Conceptualization:** Craig Wesley Carpenter, Charles M. Tolbert.

**Data curation:** Craig Wesley Carpenter, Michael C. Lotspeich-Yadao.

**Formal analysis:** Craig Wesley Carpenter, Michael C. Lotspeich-Yadao.

**Funding acquisition:** Craig Wesley Carpenter, Charles M. Tolbert.

**Investigation:** Craig Wesley Carpenter, Michael C. Lotspeich-Yadao.

**Methodology:** Craig Wesley Carpenter, Michael C. Lotspeich-Yadao.

**Project administration:** Craig Wesley Carpenter.

**Supervision:** Charles M. Tolbert.

**Visualization:** Craig Wesley Carpenter, Michael C. Lotspeich-Yadao.

**Writing – original draft:** Craig Wesley Carpenter.

**Writing – review & editing:** Craig Wesley Carpenter, Michael C. Lotspeich-Yadao, Charles M. Tolbert.

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
