## [Decision Letter · Decision Letter 0]

21 Dec 2021

PONE-D-21-36293When to Use Commuting Zones? An Empirical Description of Spatial Spillover Effects in U.S. Counties versus Commuting ZonesPLOS ONE

Dear Dr. Carpenter,

Thank you for submitting your manuscript to PLOS ONE. After careful consideration, we feel that it has merit but does not fully meet PLOS ONE’s publication criteria as it currently stands. Therefore, we invite you to submit a revised version of the manuscript that addresses the points raised during the review process.

Reviewer 1 recommended reject while Reviewer 2 recommended major revision. Both of them pointed out drawbacks in literature review and insufficient descriptions. I agree to their comments. I expect the authors to improve the manuscript by addressing the comments from them.

We look forward to receiving your revised manuscript.

Kind regards,

Hironori Kato, Dr. Eng.

Academic Editor

PLOS ONE

Journal Requirements:

[NO authors have competing interests]. 

3.  We note that Figures 1, B1-B5 in your submission contain map/satellite images which may be copyrighted. All PLOS content is published under the Creative Commons Attribution License (CC BY 4.0), which means that the manuscript, images, and Supporting Information files will be freely available online, and any third party is permitted to access, download, copy, distribute, and use these materials in any way, even commercially, with proper attribution. For these reasons, we cannot publish previously copyrighted maps or satellite images created using proprietary data, such as Google software (Google Maps, Street View, and Earth). For more information, see our copyright guidelines: http://journals.plos.org/plosone/s/licenses-and-copyright.

a) You may seek permission from the original copyright holder of Figures 1, B1-B5 to publish the content specifically under the CC BY 4.0 license.  

Reviewers' comments:

Reviewer's Responses to Questions

**Comments to the Author**

1. Is the manuscript technically sound, and do the data support the conclusions?

Reviewer #1: No

Reviewer #2: Partly

2. Has the statistical analysis been performed appropriately and rigorously? 

Reviewer #1: No

Reviewer #2: Yes

3. Have the authors made all data underlying the findings in their manuscript fully available?

Reviewer #1: Yes

Reviewer #2: Yes

4. Is the manuscript presented in an intelligible fashion and written in standard English?

Reviewer #1: Yes

Reviewer #2: Yes

5. Review Comments to the Author

Reviewer #1: Review report on PONE-D-21-36293

Overview:

This paper calculated Local Moran’s I index for various regional variables measured at the community

or commuting zone (CZ) level. They found that CZs reduce – but do not eliminate – spatial dependence

for most variables across categories. I have some major concerns about this paper, those listed as

follows.

Major concerns:

1) I think this paper is not up to the standard of international journals. The results are presented in

just 16 lines of text and there is no discussion of the results. The explanation is poor and for

instance, the definition of “High-High cluster” or “High-Low outlier” are not explained although

they are main results.

2) Table 2.1 shows the result of calculating Moran’s I-index. The value is too high. The original

Moran’s I-index takes values from [-1 to 1] when W is row-standardized. Therefore, I believe that

the values shown in this table are the standardized Moran’s I-index (z-values). However, this value

is too high compared to the sample size (in fact, the sample size itself is not clear). Since there is

no clear explanation, I could not understand what this value represents. In the explanation under

Table 2.1, you explained that it was a chi-square value, but usually, normal distribution

permutation test is used for the inference of Moran’s I. See Anselin (1995).

3) It is well known in the MAUP literature that the results of Moran’s I depend on spatial scale. Do

the results in your paper add new evidence to the existing knowledge base? I think literature

review may be improved by reflecting such perspective.

4) This paper calculated Moran’s I index at different spatial scales. However, I could not understand

what you wanted to discuss in this comparison. For example, suppose we have a situation where

a CZ is classified as High-Low, but a community belonging to the CZ is classified as High-High.

How should the readers of your paper interpret such a result?

Specific comments

1. P.5 L.91: may be ®. Please confirm the definition of Moran’s I.

2. P.5 L.96: Moran’s I does not presuppose homogeneity. The terms homogeneity, heterogeneity

and spillover in this paper are vague and difficult to understand.

3. P.5 L.105 Anselin (1993) may be Anselin (1996).

4. Please add descriptive statistics.

5. Please add the definition of significance code (for instance, *** in table 2.1).

6. Please be considerate of non-US readers.

Reviewer #2: Comment 1:

The authors should explicitly state their contribution to the literature if there is already a research article that examined the property and problem when using CZs in empirical analysis. For example, Foote et al. (2021), the article I cited below, also examined this sort of problem and showed the guideline for empirical researchers.

Comment 2:

I suggest that the authors check if the robustness of their main results (CZs reduce spatial autocorrelation more compared to the aggregation using counties) using the alternative global indicator of spatial autocorrelation (GISA) like Geary’s C. As pointed out in, for example, Mendes & Mendes (2015), Geary’s C is a GISA more sensitive to local autocorrelation while Global Moran’s I can capture global one. I’m wondering if this difference in property does not alter the main conclusion.

Comment 3:

I also recommend the authors report the results using different cut-off values of tree height in clustering and examine if this difference does not alter their conclusion, following the guideline proposed by Foote et al. (2021), for example. This sort of exercise provides supportive evidence that the main results are consistent regardless of cut-off.

Foote, A., Kutzbach, M. J., & Vilhuber, L. (2021). Recalculating...: How Uncertainty in Local Labour Market Definitions Affects Empirical Findings. Applied Economics, 53(14), 1598-1612.

Mendes, D. A., & Mendes, V. M. (2015). Parametric models in spatial econometrics: a survey. In Complexity and Geographical Economics (pp. 51-71). Springer, Cham.

6. PLOS authors have the option to publish the peer review history of their article (what does this mean?). If published, this will include your full peer review and any attached files.

Reviewer #1: No

Reviewer #2: No

---

## [Author Response · Author response to Decision Letter 0]

31 Mar 2022

Much of the formatting is lost in our response to reviewers when pasting into this form. We suggest using the uploaded Word document "Response to Reviewers" (which contains identical text without loss of formatting) for easier reading. Thank you.

When to Use U.S. Commuting Zones? An Empirical Description of Spatial Autocorrelation in U.S. Counties versus Commuting Zones

Response to Reviewers

Additional Comments from Editors (3/30/2022)

Thank you for obtaining copyright permission from the original copyright holder of the Commuting Zones shapefile. As a final step, in the Figure 1 caption, please include the following text: “Republished from [ref] under a CC BY license, with permission from [name of publisher], original copyright [original copyright year].”

 We have added the requested caption to figure 1.

 We again note that the shapefiles themselves, which is what we use to create our own maps, are not copyrighted. But we added the caption, regardless.

Additional Comments from Editors (3/16/2022)

We've checked your submission and before we can proceed, we need you to address the following issues:

1. Thank you for providing the following response to our previous copyright query: "• Figures 1, as well as B1-B5 does not contain any map or satellite imagery that is copyrighted. These figures were generated from publicly available data, and copyright protection is not available for any work of the United States Government (Title 17 U.S.C., Section 105).

• A reference to this was added to section ‘Data’. “The Census 2000 shapefile was obtained from the U.S. Census Bureau to represent county-level geography (U.S. Census Bureau 2019) and the Commuting Zones shapefile from Fowler and Jensen (2020)” Corresponding citations for these data sources are now included in the bibliography."

Please note that the Fowler and Jensen (2020) work, while free to view online, is indeed copyrighted. SagePub permissions policy (https://urldefense.com/v3/__https://us.sagepub.com/en-us/nam/journals-permissions__;!!KwNVnqRv!SceAhBSxNYZGn25eA-OR1hsGf1zThpqwwwQfIofIQ1cprzthjevdUsxzbH2YOhKtRg$ ) states the following: "In some instances, SAGE may make its content freely viewable; however, such material may require permission for reuse."

SagePub does have its own process for requesting permissions (https://urldefense.com/v3/__https://us.sagepub.com/en-us/nam/process-for-requesting-permission__;!!KwNVnqRv!SceAhBSxNYZGn25eA-OR1hsGf1zThpqwwwQfIofIQ1cprzthjevdUsxzbH2d0ta9HA$ ), however this process mostly relies on RightsLink. Please note that RightsLink permission forms often impose use restrictions that are incompatible with our CC BY 4.0 license, and we are therefore unable to accept these permissions. For this reason, we strongly recommend contacting copyright holders with the PLOS ONE Request for Permission form.

To seek permission from the copyright owner to publish your map figures under the specific Creative Commons Attribution License (CCAL), CC BY 4.0, please contact them with the following text and PLOS ONE Request for Permission form (https://urldefense.com/v3/__http://journals.plos.org/plosone/s/file?id=7c09*content-permission-form.pdf__;Lw!!KwNVnqRv!SceAhBSxNYZGn25eA-OR1hsGf1zThpqwwwQfIofIQ1cprzthjevdUsxzbH0UJWcNWA$ ):

“I request permission for the open-access journal PLOS ONE to publish XXX under the Creative Commons Attribution License (CCAL) CC BY 4.0 (https://urldefense.com/v3/__http://creativecommons.org/licenses/by/4.0/__;!!KwNVnqRv!SceAhBSxNYZGn25eA-OR1hsGf1zThpqwwwQfIofIQ1cprzthjevdUsxzbH0ze3EBpA$ ). Please be aware that this license allows unrestricted use and distribution, even commercially, by third parties. Please reply and provide explicit written permission to publish XXX under a CC BY license.”

Please upload the granted permission to the manuscript as a Supporting Information file. In the figure caption of the copyrighted figure, please include the following text: “Republished from [ref] under a CC BY license, with permission from [name of publisher], original copyright [original copyright year].”

If you are unable to obtain permission from the copyright holder, please either A) remove the figure or B) supply a replacement figure that complies with the CC BY 4.0 license. Please check copyright information on all replacement figures and update the figure caption with source information.

USGS National Map Viewer (https://urldefense.com/v3/__http://viewer.nationalmap.gov/viewer/__;!!KwNVnqRv!SceAhBSxNYZGn25eA-OR1hsGf1zThpqwwwQfIofIQ1cprzthjevdUsxzbH2pY73nIw$ )

USGS Earth Resources Observatory and Science (EROS) Center (https://urldefense.com/v3/__http://eros.usgs.gov/*__;Iw!!KwNVnqRv!SceAhBSxNYZGn25eA-OR1hsGf1zThpqwwwQfIofIQ1cprzthjevdUsxzbH0mpob9Hg$ )

The Gateway to Astronaut Photography of Earth (https://urldefense.com/v3/__https://eol.jsc.nasa.gov/__;!!KwNVnqRv!SceAhBSxNYZGn25eA-OR1hsGf1zThpqwwwQfIofIQ1cprzthjevdUsxzbH0ZH3oOpQ$ )

Maps at the CIA (https://urldefense.com/v3/__https://www.cia.gov/library/publications/the-world-factbook/docs/refmaps.html__;!!KwNVnqRv!SceAhBSxNYZGn25eA-OR1hsGf1zThpqwwwQfIofIQ1cprzthjevdUsxzbH0UA8m2aQ$ )

NASA Earth Observatory (https://urldefense.com/v3/__http://earthobservatory.nasa.gov/__;!!KwNVnqRv!SceAhBSxNYZGn25eA-OR1hsGf1zThpqwwwQfIofIQ1cprzthjevdUsxzbH1M5d8kXw$ )

Landsat (https://urldefense.com/v3/__http://landsat.visibleearth.nasa.gov/__;!!KwNVnqRv!SceAhBSxNYZGn25eA-OR1hsGf1zThpqwwwQfIofIQ1cprzthjevdUsxzbH2PfC-O5Q$ )

Natural Earth (https://urldefense.com/v3/__http://www.naturalearthdata.com/__;!!KwNVnqRv!SceAhBSxNYZGn25eA-OR1hsGf1zThpqwwwQfIofIQ1cprzthjevdUsxzbH3wo9_VzQ$ )

We've returned your manuscript to your account. Please resolve these issues and resubmit your manuscript within 21 days. If you need more time, please email the journal office at plosone@plos.org. We are happy to grant extensions of up to one month past this due date. If we do not hear from you within 21 days, we will withdraw your manuscript.

Please log on to PLOS Editorial Manager at https://www.editorialmanager.com/pone/ to access your manuscript. You will find your manuscript in the 'Submissions Sent Back to Author' link under the New Submissions menu. Be sure to remove your previous manuscript file if you are uploading a new file in response to these requests. After you've made the changes requested above, please be sure to view and approve the revised PDF after rebuilding the PDF to complete the resubmission process.

We are requesting these changes to comply with the PLOS ONE submission guidelines (https://urldefense.com/v3/__https://journals.plos.org/plosone/s/submission-guidelines__;!!KwNVnqRv!SceAhBSxNYZGn25eA-OR1hsGf1zThpqwwwQfIofIQ1cprzthjevdUsxzbH1fPR9jqw$ ). Please note that we won't send your manuscript for review until you have resolved the above requests. 

Thank you for submitting your work to PLOS ONE and supporting our mission of Open Science.

 In response to the copywrite concerns (emailed to the authors 3/16/2022), we have retrieved a signed Content Permission Form.

 We note that we do not agree that this form is necessary. We used replication data (a shapefile) available from Dr. Chris Fowler (https://sites.psu.edu/psucz/). Dr. Fowler strongly disagrees with the idea that Sage has any ownership of Dr. Fowler’s data because they published his article. While they may have rights to the images made with that data in that article, but they do not own the data itself. We use the shapefile/data to make our own maps. This is not covered by Sage’s copyright.

 Regardless, Dr. Fowler agreed to sign the Content Permission Form, if the editors still believe it is necessary. And we attached the form to the new resubmission.

Editor Comments

Thank you for submitting your manuscript to PLOS ONE. After careful consideration, we feel that it has merit but does not fully meet PLOS ONE’s publication criteria as it currently stands. Therefore, we invite you to submit a revised version of the manuscript that addresses the points raised during the review process.

Reviewer 1 recommended reject while Reviewer 2 recommended major revision. Both of them pointed out drawbacks in literature review and insufficient descriptions. I agree to their comments. I expect the authors to improve the manuscript by addressing the comments from them.

Thank you for the opportunity to revise and resubmit our article. We are grateful for your guidance and the reviewers’ the detailed and specific comments. We paste the reviewer and technical comments below, bulleting our respective responses and edits below each comment. We find the article much improved as a result of these changes.

 

Reviewer Comments

Reviewer #1

Overview:

This paper calculated Local Moran’s I index for various regional variables measured at the community or commuting zone (CZ) level. They found that CZs reduce – but do not eliminate – spatial dependence for most variables across categories. I have some major concerns about this paper, those listed as follows.

Major concerns:

 I think this paper is not up to the standard of international journals. The results are presented in just 16 lines of text and there is no discussion of the results. The explanation is poor and for instance, the definition of “High-High cluster” or “High-Low outlier” are not explained although they are main results.

 Although we attempt to keep the article concise in nature, we take this comment seriously. We add explanations/definitions of the clusters, as well as related citations that have used this framework previously at the end of the Reginal Variation in Spatial Autocorrelation subsection. We also expanded the explanation and discussion in the results section, elaborating on the comparison of the clusters between counties and CZs.

 Table 2.1 shows the result of calculating Moran’s I-index. The value is too high. The original Moran’s I-index takes values from [-1 to 1] when W is row-standardized. Therefore, I believe that the values shown in this table are the standardized Moran’s I-index (z-values). However, this value is too high compared to the sample size (in fact, the sample size itself is not clear). Since there is no clear explanation, I could not understand what this value represents. In the explanation under Table 2.1, you explained that it was a chi-square value, but usually, normal distribution permutation test is used for the inference of Moran’s I. See Anselin (1995).

 We thank the reviewer for this comment. We have replaced the values with [-1, 1] values and use the suggested usual method for inference (significance).

 It is well known in the MAUP literature that the results of Moran’s I depend on spatial scale. Do the results in your paper add new evidence to the existing knowledge base? I think literature review may be improved by reflecting such perspective.

 This is a valuable comment and overlaps with some comments from reviewer 2 regarding how expanding the literature review of MAUP will enhance the article. In particular, we agree that noting the relationship of this article to the MAUP literature will help clarify our contribution.

 Specifically, we note important contributions to MAUP and how they relate to CZ delineations more generally, while also noting this article focuses on the specific US CZ delineations commonly used by researchers.

 This paper calculated Moran’s I index at different spatial scales. However, I could not understand what you wanted to discuss in this comparison. For example, suppose we have a situation where a CZ is classified as High-Low, but a community belonging to the CZ is classified as High-High. How should the readers of your paper interpret such a result?

 We have expanded the discussion of how to compare the maps across the spatial scales. Following the reviewer’s suggestion, we specifically give some more discussion to the results (in the results section):

 “To demonstrate, we use the Moran’s I (local) statistic and LISA to visualize the potential for regional variation in spatial autocorrelation in the appendix. Example results, presented in Figure 1, indicate that results are not generally driven by regional outliers and rather, the spatial unit selected can alter the statistical significance of the clusters. When taking into consideration manufacturing, for instance, low concentration clusters appear when using CZs but disappear from the west coast when using counties as the spatial unit. For the percent of residents identifying as Black, Appalachia as a whole is a Low-Low cluster at the county-level but a Low-High outlier to neighboring units when CZs are used. This is a clear example of the MAUP, as the statistical significance of the Moran’s I (local) statistic can change based on the spatial units used. While CZs may change local regional concentration of spatial autocorrelation, larger regional variation found with counties is not typically eliminated. Full results are available in the appendix.”

Specific comments:

 P.5 L.91: x_i may be (x_i-x ® ). Please confirm the definition of Moran’s I.

 This is a great catch. We have fixed the definition.

 P.5 L.96: Moran’s I does not presuppose homogeneity. The terms homogeneity, heterogeneity

and spillover in this paper are vague and difficult to understand.

 This is also a good catch. We drop the reference to homogeneity here as it relates to Moran’s I.

 We also (1) clarify the earlier reference to heterogeneity by noting that it refers to heterogeneity in commuting patterns across CZ boundaries, (2) provide a reference, and (3) provide footnote 6 elaborating on that reference to help explain the extent of heterogeneity in commuting across CZ boundaries.

 Finally, regarding the use of “spillover,” we agree with the reviewer that this is a less commonly used term in the literature and our use was unclear. We replaced every instance of “spillover” and “spillover effects” to now more clearly refer to “spatial autocorrelation”, including in the title of the article. We thank the reviewer for this suggestion and agree that the clarity has improved.

 P.5 L.105 Anselin (1993) may be Anselin (1996).

 This is correct; Anselin (1993) should be Anselin (1996). This is based on what is already in the bibliography and we have corrected this in-text error.

 Please add descriptive statistics.

 Although descriptive statistics of the values of the variables themselves are not necessarily essential to an empirical description of their spatial autocorrelation, we agree with the review that descriptive statistics may be of interest, especially as they enhance readers’ understanding of the variables and enhancing the understanding of how the descriptive statistics vary between counties and CZs. We added descriptive statistics to the appendix.

 Please add the definition of significance code (for instance, *** in table 2.1).

 We added the definition to the notes of each table with significance levels: “Significance levels: ***<1%, **<5%, *<10%”.

 Please be considerate of non-US readers.

 We take this comment seriously, especially as many non-US scientists use CZ or county-level US data.

 First, we create a literature review section on MAUP, paying particular attention to where non-US readers may benefit from some additional discussion of US geography.

 This new section includes particular attention to the broader context:

 Cliff, Andrew D., and Keith Ord. 1973. Spatial Autocorrelation. London: Pion.

 Cliff, Andrew D., and Keith Ord. 1981. Spatial Processes: Models and Applications. London: Taylor & Francis.

 Openshaw, Stan. 1983. “The modifiable areal unit problem.” Concepts and Techniques in Modern Geography, 38. Norwich: Geobooks.

 Fotheringham, A. Stewart, and David WS Wong. 1991. "The modifiable areal unit problem in multivariate statistical analysis." Environment and Planning A (23) 7: 1025-1044.

 Nelson, Jonathan K., and Cynthia A. Brewer. 2017. "Evaluating data stability in aggregation structures across spatial scales: revisiting the modifiable areal unit problem." Cartography and Geographic Information Science 44 (1): 35-50.

 Aron, Joan L., Sheryl Luzzadder-Beach, Min Sun, Fahui Wang, Laura E. Jackson, Greg Susanke, Edward Washburn. White, J., Wong, D., Yang, C. and Young, S. 2010. “Environmental and related applications.” Advanced Geoinformation Science, 303-50.

 Lery, Bridgette. 2008. "A comparison of foster care entry risk at three spatial scales." Substance Use & Misuse 43 (2): 223-237.

 Swift, Andrew, Lin Liu, and James Uber. 2014. "MAUP sensitivity analysis of ecological bias in health studies." GeoJournal 79(2): 137-153.

 Lee, Dong Wook, and Melissa Rogers. 2019. "Measuring geographic distribution for political research." Political Analysis 27(3): 263-280.

 Chi, Guangqing and Jun Zhu. 2020. Spatial Regression Models for the Social Sciences. SAGE Publications: Thousand Oaks, CA.

 Griffith, Daniel A. 1987. Spatial Autocorrelation: A Primer. Association of American Geographers: Washington, DC.

 Nielsen, Michael Meinild, and Pontus Hennerdal. 2017. "Changes in the residential segregation of immigrants in Sweden from 1990 to 2012: Using a multi-scalar segregation measure that accounts for the modifiable areal unit problem." Applied Geography 87: 73-84.

 Serra, Miguel, Sophia Psarra, and Jamie O’Brien. 2018. "Social and physical characterization of urban contexts: Techniques and methods for quantification, classification and purposive sampling." Urban Planning 3(1): 58-74.

 Second, we clarify the relevance of our findings to non-US researchers in the conclusion by citing a number of examples of researchers applied Tolbert and Sizer’s (1996) work to non-US contexts. 

We thank reviewer 1 for their specific and constructive comments. We find the article much improved as a result of these changes.

 

Reviewer #2

 The authors should explicitly state their contribution to the literature if there is already a research article that examined the property and problem when using CZs in empirical analysis. For example, Foote et al. (2021), the article I cited below, also examined this sort of problem and showed the guideline for empirical researchers.

 This comment and overlaps with a comment from Reviewer #1 that expanding the literature review to better position this article in the related literature would be valuable and help clarify this article’s contribution. Pursuantly, after discussing the more general literature on MAUP, we also add a discussion of Foote et al. (2021) and how this article relates thereto. 

 I suggest that the authors check if the robustness of their main results (CZs reduce spatial autocorrelation more compared to the aggregation using counties) using the alternative global indicator of spatial autocorrelation (GISA) like Geary’s C. As pointed out in, for example, Mendes & Mendes (2015), Geary’s C is a GISA more sensitive to local autocorrelation while Global Moran’s I can capture global one. I’m wondering if this difference in property does not alter the main conclusion.

 This is an important comment and a valuable check on our results. We add a new discussion in our methods section on Geary’s C discussing its higher sensitivity to local spatial autocorrelation (relative to Moran’s I) and reproduce the estimates of spatial autocorrelation using Geary’s C, citing Mendes & Mendes (2015). 

 We then produce Geary’s C estimates for our variables in a new table 3 in the results section. In general, we find that results are consistent with the those found when using Moran’s I. Nonetheless, we think that this is a valuable exercise and increases the robustness of our results and thank the reviewer for this suggestion.

 I also recommend the authors report the results using different cut-off values of tree height in clustering and examine if this difference does not alter their conclusion, following the guideline proposed by Foote et al. (2021), for example. This sort of exercise provides supportive evidence that the main results are consistent regardless of cut-off.

 We think that this is an important point resulting from (as this reviewer also points out) insufficiently contextualizing this article on the related literature. To help address this point, we (1) created a MAUP literature review section that discusses the relevance of Foote et al.’s (2021) important work and how it related to this article; (2) added a discussion in the conclusion section of Foote et al.’s (2021) ongoing relevance to researchers deciding to use the US commuting zone definition; (3) changed the first sentence of the abstract to emphasize that we are examining the commonly used U.S. CZ delineations; and (4) make a note in the conclusions that we are not arguing external validity of our findings to all delineations of labor market areas but rather focusing on the commonly used U.S. CZ delineations (though we acknowledge the guide may still be useful there as a rule of thumb for researchers). 

 Foote, A., Kutzbach, M. J., & Vilhuber, L. (2021). Recalculating...: How Uncertainty in Local Labour Market Definitions Affects Empirical Findings. Applied Economics, 53(14), 1598-1612.

Mendes, D. A., & Mendes, V. M. (2015). Parametric models in spatial econometrics: a survey. In Complexity and Geographical Economics (pp. 51-71). Springer, Cham.

 We thank the reviewer for suggesting these excellent citations. We have added them to the article accompanying their respective discussions.

We similarly thank reviewer 2 for their specific comments and the suggested citations. We now find the article better (and more explicitly) situated in the recent related literature.

 

Technical Comments

Original Revision Comments:

We note that Figures 1, B1-B5 in your submission contain map/satellite images which may be copyrighted. All PLOS content is published under the Creative Commons Attribution License (CC BY 4.0), which means that the manuscript, images, and Supporting Information files will be freely available online, and any third party is permitted to access, download, copy, distribute, and use these materials in any way, even commercially, with proper attribution. For these reasons, we cannot publish previously copyrighted maps or satellite images created using proprietary data, such as Google software (Google Maps, Street View, and Earth). For more information, see our copyright guidelines: http://journals.plos.org/plosone/s/licenses-and-copyright.

 a) You may seek permission from the original copyright holder of Figures 1, B1-B5 to publish the content specifically under the CC BY 4.0 license. 

 Figures 1, as well as B1-B5 does not contain any map or satellite imagery that is copyrighted. These figures were generated from publicly available data, and copyright protection is not available for any work of the United States Government (Title 17 U.S.C., Section 105).

 A reference to this was added to section ‘Data’. “The Census 2000 shapefile was obtained from the U.S. Census Bureau to represent county-level geography (U.S. Census Bureau 2019) and the Commuting Zones shapefile from Fowler and Jensen (2020)” Corresponding citations for these data sources are now included in the bibliography.

---

## [Decision Letter · Decision Letter 1]

20 Apr 2022

PONE-D-21-36293R1When to Use Commuting Zones? An Empirical Description of Spatial Autocorrelation in U.S. Counties versus Commuting ZonesPLOS ONE

Dear Dr. Carpenter,

Thank you for submitting your revised manuscript to PLOS ONE. After careful consideration, we feel that it has merit but does not fully meet PLOS ONE’s publication criteria as it currently stands. Therefore, we invite you to submit a revised version of the manuscript that addresses the points raised during the review process.

Reviewer 2 raised questions on a methodological issue while the reviewer also pointed out that the results do not well support the conclusion. I expect the authors to improve the manuscript by addressing the comments.

We look forward to receiving your revised manuscript.

Kind regards,

Hironori Kato, Dr. Eng.

Academic Editor

PLOS ONE

Journal Requirements:

Reviewers' comments:

Reviewer's Responses to Questions

**Comments to the Author**

1. If the authors have adequately addressed your comments raised in a previous round of review and you feel that this manuscript is now acceptable for publication, you may indicate that here to bypass the “Comments to the Author” section, enter your conflict of interest statement in the “Confidential to Editor” section, and submit your "Accept" recommendation.

Reviewer #1: All comments have been addressed

Reviewer #2: (No Response)

2. Is the manuscript technically sound, and do the data support the conclusions?

Reviewer #1: Partly

Reviewer #2: No

3. Has the statistical analysis been performed appropriately and rigorously? 

Reviewer #1: Yes

Reviewer #2: I Don't Know

4. Have the authors made all data underlying the findings in their manuscript fully available?

Reviewer #1: Yes

Reviewer #2: Yes

5. Is the manuscript presented in an intelligible fashion and written in standard English?

Reviewer #1: Yes

Reviewer #2: Yes

6. Review Comments to the Author

Reviewer #1: Although I suggested the rejection at the first stage, you responded well to my concerns. Therefore, I have no further comments.

Reviewer #2: I confirmed the revised version of the manuscript and could find that my suggestions were largely reflected.

However, I have the following concerns or questions after reading the revised manuscript.

Comment 1:

The authors wrote that they used the Moran test for spatial dependence (chi-squared) when examining the statistical significance of the calculated values of local Moran's I and Geary's C in the footnotes of the tables (e.g., Table 2.1). However, as Reviewer #1 also pointed out, the statistical test for these local indices generally relies on Z-test rather than the chi-squared test (at least, I have never seen the use of the chi-squared test in any article).

Since there is no information on, for example, the formulation of test statistics, I could not judge if the obtained results were certain.

Although the authors replied that they have replaced the values with [-1, 1] values and used the suggested usual method for inference (significance), I am not sure if the correction was conducted properly as far as I checked the footnotes for the tables.

Also, the values of test statistics or standard error in addition to calculated local Moran's and Geary's C should be provided. Only showing significance code does not provide sufficient information, I think.

(By the way, it seems that the authors used the sentence "Table summarizes the significance of the Moran test for spatial dependence" for the results of local Geary's C by mistake in Table 3.1 and Table 3.2, for example.)

Comment 2:

The authors concluded that CZs do indeed reduce the severity of spatial autocorrelation. However, as far as I checked the tables in the main text, I am not sure if this story consistently holds. For several ecological variables and LQs, the values of local Moran's I and Geary's C for CZs are larger than those for counties. And even if the local indices for CZs are smaller than those for counties, it is difficult to judge whether the difference between the local indices for counties and that for CZs is notable.

I suggest that the authors reconsider the conclusion (and thus the structure of their manuscript) if they cannot provide a clear explanation for these matters.

7. PLOS authors have the option to publish the peer review history of their article (what does this mean?). If published, this will include your full peer review and any attached files.

Reviewer #1: No

Reviewer #2: No

---

## [Author Response · Author response to Decision Letter 1]

2 Jun 2022

Much of the formatting is lost in our response to reviewers when pasting here. We suggest using the uploaded Word document "Response to Reviewers" (which contains identical text without loss of formatting) for easier reading. Thank you.

Response to Reviewers

Editor Comments

Thank you for submitting your revised manuscript to PLOS ONE. After careful consideration, we feel that it has merit but does not fully meet PLOS ONE’s publication criteria as it currently stands. Therefore, we invite you to submit a revised version of the manuscript that addresses the points raised during the review process.

Reviewer 2 raised questions on a methodological issue while the reviewer also pointed out that the results do not well support the conclusion. I expect the authors to improve the manuscript by addressing the comments.

Thank you for the opportunity to revise and resubmit our article. We are grateful for your guidance and the reviewers’ the detailed and specific comments. We paste the reviewer and technical comments below, bulleting our respective responses and edits below each comment. We find the article improved as a result of these changes.

 

Reviewer Comments

Reviewer #1

1. All comments have been addressed.

2. Although I suggested the rejection at the first stage, you responded well to my concerns. Therefore, I have no further comments.

• We reiterate our thanks to this reviewer for reviewing our article again, and for their specific and constructive comments in the previous round of reviews. We find the article much improved as a result.

Reviewer #2

1. I confirmed the revised version of the manuscript and could find that my suggestions were largely reflected. However, I have the following concerns or questions after reading the revised manuscript. The authors wrote that they used the Moran test for spatial dependence (chi-squared) when examining the statistical significance of the calculated values of local Moran's I and Geary's C in the footnotes of the tables (e.g., Table 2.1). However, as Reviewer #1 also pointed out, the statistical test for these local indices generally relies on Z-test rather than the chi-squared test (at least, I have never seen the use of the chi-squared test in any article). 

• We thank the reviewer for their previous comments on this article.

• ‘Chi-squared’, as indicated in the footnote of the tables, was a typographic error. We updated all table footnotes to indicate that the table summarizes a test for spatial dependence or spatial autocorrelation, with significance based on the z-score.

2. Since there is no information on, for example, the formulation of test statistics, I could not judge if the obtained results were certain. Although the authors replied that they have replaced the values with [-1, 1] values and used the suggested usual method for inference (significance), I am not sure if the correction was conducted properly as far as I checked the footnotes for the tables.

• We added a sentence in the table footnotes with more detail on the statistical program used to make these calculations. We indicate that “the weight was constructed in GeoDa 1.16, and test performed with the SPATGSA (Pisati, 2001) command in Stata 17/SE.”

• Further, we indicate every time that that Global Geary’s c test is used that the contiguity ratio ranges from 0 to 2.

3. Also, the values of test statistics or standard error in addition to calculated local Moran's and Geary's C should be provided. Only showing significance code does not provide sufficient information, I think.

• Following the reviewer’s suggestion, we added standard errors and z-scores to these tables.

i. The original package (SpaceStat) did not support the production of standard error or z-scores for the Global Geary’s c test. To ensure homogeneity throughout the paper and ease replicability, we retained the weights and replaced SpaceStat with an ado package (SPATGSA) that provided test statistics and standard error for both the Global Moran’s I and Global Geary’s c. While some variation in the output has arisen, it does not change the outcome of the paper.

ii. We were also reminded through this process that contiguity-based spatial weighting matrices were standard in the literature for county-level analysis and switched these tables with the inverse-distance based tables in the Appendix.

4. (By the way, it seems that the authors used the sentence "Table summarizes the significance of the Moran test for spatial dependence" for the results of local Geary's C by mistake in Table 3.1 and Table 3.2, for example.)

• We thank the reviewer for catching this typographical error. We have fixed it.

5. The authors concluded that CZs do indeed reduce the severity of spatial autocorrelation. However, as far as I checked the tables in the main text, I am not sure if this story consistently holds. For several ecological variables and LQs, the values of local Moran's I and Geary's C for CZs are larger than those for counties. And even if the local indices for CZs are smaller than those for counties, it is difficult to judge whether the difference between the local indices for counties and that for CZs is notable. I suggest that the authors reconsider the conclusion (and thus the structure of their manuscript) if they cannot provide a clear explanation for these matters.

• This is a valuable point by the reviewer and we are grateful for it. We agree that our previous conclusions were too broad and have changed the language (and added some specificity around variables) to tone down the generalizability of the conclusions.

• While the results do indicate that it is more likely to see reduced spatial autocorrelation when moving to CZs, there are a number of important exceptions that the reviewer correctly points out, and the reviewer is correct that in some cases, it is difficult to judge whether the difference between the indices is notable. We added these important caveats to our conclusions and abstract.

• The abstract now reads: “U.S. Commuting Zones (CZs) are an aggregation of county-level data that researchers commonly use to create less arbitrary spatial entities and to reduce spatial autocorrelation. However, by further aggregating data, researchers lose point data and the associated detail. Thus, the choice between using counties or CZs often remains subjective with insufficient empirical evidence guiding researchers in the choice. This article categorizes regional data as entrepreneurial, economic, social, demographic, or industrial and tests for the existence of local spatial autocorrelation in county and CZ data. We find CZs often reduce – but do not eliminate and can even increase – spatial autocorrelation for variables across categories. We then test the potential for regional variation in spatial autocorrelation with a series of maps and find variation based on the variable of interest. We conclude that the use of CZs does not eliminate the need to test for spatial autocorrection, but CZs may be useful for reducing spatial autocorrelation in many cases.”

---

## [Editor Report · Decision Letter 2]

8 Jun 2022

When to Use Commuting Zones? An Empirical Description of Spatial Autocorrelation in U.S. Counties versus Commuting Zones

PONE-D-21-36293R2

Dear Dr. Carpenter,

We’re pleased to inform you that your manuscript has been judged scientifically suitable for publication and will be formally accepted for publication once it meets all outstanding technical requirements.

Kind regards,

Hironori Kato, Dr. Eng.

Academic Editor

PLOS ONE

Reviewers' comments:

Thank you for your patience.

---

## [Editor Report · Acceptance letter]

22 Jun 2022

PONE-D-21-36293R2 

When to Use Commuting Zones? An Empirical Description of Spatial Autocorrelation in U.S. Counties versus Commuting Zones 

Dear Dr. Carpenter:

I'm pleased to inform you that your manuscript has been deemed suitable for publication in PLOS ONE. Congratulations! Your manuscript is now with our production department. 

Kind regards, 

on behalf of

Dr. Hironori Kato 

Academic Editor

PLOS ONE